# DBGMS: A Dual-Branch Generative Adversarial Network with Multi-Task Self-Supervised Enhancement for Robust Auditory Attention Decoding

## Abstract

Detecting auditory attention from brain signals has been a significant challenge in neuroscience and brain-computer interface research. While progress has been made in EEG-based auditory attention detection, existing methods often struggle with limited data and short decision windows, particularly in complex auditory environments. In this paper, we propose DBGMS (Dual-Branch Generative Adversarial Network with Multi-Task Self-Supervised Enhancement), a novel framework for robust auditory attention decoding from electroencephalogram (EEG) signals. There are three key innovations in our approach: (1) A dual-branch architecture is developed that combines temporal attention and frequency residual learning, enabling more comprehensive feature extraction to be achieved from EEG signals; (2) Branch-specific generative adversarial networks (GANs) are designed to generate high-quality augmented samples in both temporal and frequency domains, effectively addressing the data scarcity issue in auditory attention decoding; (3) Attention mechanisms and graph convolution operations are incorporated in both temporal and frequency domains. (4) A multi-task self-supervised learning strategy is introduced, incorporating several complementary tasks such as temporal order prediction, frequency band reconstruction, and time-frequency consistency. This approach leverages unlabeled data to enhance the model's ability to capture subtle attention-related features from multiple perspectives, thereby improving generalization across subjects and listening conditions. In contrast to state-of-the-art methods, DBGMS presents significant improvements in detection accuracy and robustness, particularly for short decision windows. Our framework is evaluated on two public EEG datasets, including KUL and DTU, demonstrating its effectiveness across various experimental settings.

## 1 Introduction

Decoding auditory attention from electroencephalogram (EEG) signals presents significant challenges in neuroscience and brain-computer interface researchBassett & Sporns (2017); Britton et al. (2016). While existing methods often struggle with limited data and short decision windows, particularly in complex auditory environmentsCherry (1953); Choi et al. (2013). The inherent variability of EEG signals, stemming from inter-subject differences, intra-subject variations, and environmental factors, further complicates this taskLuck (2014); Blankertz et al. (2016); O'Sullivan et al. (2015c).

Recent advancements in deep learning and signal processing have driven substantial improvements in auditory attention decoding. Convolutional Neural Networks (CNNs) and Recurrent Neural Networks (RNNs) have demonstrated remarkable success, significantly outperforming traditional linear methodsDeckers et al. (2021); Ciccarelli et al. (2019); O'Sullivan et al. (2015b). Concurrently, advanced signal processing techniques such as Independent Component Analysis (ICA) and Common Spatial Patterns (CSP) have enhanced EEG signal quality and feature extractionDas et al. (2016a); Wong et al. (2018b). Despite these advancements, the transition from controlled laboratory settings to real-world applications reveals persistent challenges. Significant declines in decoding accuracy are observed when moving from controlled to naturalistic listening conditionsO'Sullivan et al.

(2017a); Fuglsang et al. (2017a). To address these issues, various methodological approaches have been explored, including adaptive filtering techniques, transfer learning methods, and robust feature extraction algorithmsO'Sullivan et al. (2017b); Fuglsang et al. (2017b). These techniques aim to mitigate the effects of inter-subject and intra-subject variability, enhancing model generalization across different subjects and recording sessions.

The challenge is further compounded by the scarcity of labeled EEG data for auditory attention tasks, which limits the diversity and representativeness of training datasetsDas et al. (2016b); Wong et al. (2018a). This limitation not only constrains the development of robust and generalizable models but also raises questions about how effectively these models can capture the true diversity of EEG patterns across a broader population. Moreover, practical applications demand real-time processing capabilities, which often leads to reduced accuracy when decision windows are shortenedAlickovic et al. (2019a); Miran et al. (2018).

To address these challenges, innovative approaches have begun to be explored. Generative Adversarial Networks (GANs) have shown potential for EEG data augmentation, effectively replicating various sources of EEG variabilityHartmann et al. (2018); Li et al. (2021); Abdelfattah et al. (2021). Self-supervised learning techniques have demonstrated promise in capturing the complex, non-stationary nature of EEG data across different subjects and recording sessionsBanville et al. (2019); Kostas et al. (2021). These approaches enable the extraction of robust features that are invariant to many sources of EEG variability, leveraging unlabeled data to improve downstream task performance.

The integration of both temporal and spectral information in EEG signals has been recognized as crucial for comprehensive auditory attention decodingDing & Simon (2012); Alickovic et al. (2019b); O'Sullivan et al. (2015a). Time-frequency analysis techniques have been employed to capture the dynamic nature of auditory attention, achieving robust performance across various experimental conditions. Additionally, graph-based approaches for EEG signal processing have shown superior performance compared to traditional architectures, leveraging the inherent spatial relationships between EEG electrodesWang et al. (2014); Zhong et al. (2020).

Despite these advancements, a unified framework that effectively combines these various techniques to address the specific challenges of auditory attention decoding has yet to be developed. Such a framework would need to address the data scarcity issue, account for inter-subject variability, maintain high accuracy with short decision windows, and effectively integrate both temporal and spectral information from EEG signals.

In this paper, DBGMS (Dual-Branch Generative Adversarial Network with Multi-Task Self-Supervised Enhancement) is proposed as a novel framework designed to address these challenges in robust auditory attention decoding from electroencephalogram (EEG) signals. This approach builds upon recent advances in deep learning and signal processing, adapting these advances to the specific requirements of auditory attention decoding.

The main contributions of our work are as follows:

- A dual-branch architecture is developed that combines temporal attention and frequency residual learning, enabling more comprehensive feature extraction from EEG signals. This approach allows for the capture of both temporal dynamics and spectral characteristics crucial for auditory attention decoding.

- Branch-specific generative adversarial networks (GANs) are designed to generate high-quality augmented samples in both temporal and frequency domains. This novel data augmentation strategy addresses both the data scarcity issue and the inherent variability of EEG signals in auditory attention decoding. By generating diverse synthetic samples, these GANs enhance the model's ability to handle varied EEG patterns, thereby improving robustness and generalization across different subjects and recording conditions.

- Attention mechanisms and graph convolution operations are incorporated in both temporal and frequency domains, enhancing the model's ability to capture relevant spatial-temporal patterns in EEG signals.

- A multi-task self-supervised learning strategy is introduced, incorporating several complementary tasks such as temporal order prediction, frequency band reconstruction, and time-frequency consistency. This approach leverages unlabeled data to enhance the model's

ability to capture subtle attention-related features from multiple perspectives, thereby improving generalization across subjects and listening conditions.

# 2 PROPOSED METHOD

The proposed Dual-Branch Generative Adversarial Network with Multi-Task Self-Supervised Enhancement (DBGMS) is designed to address the challenges in robust auditory attention decoding. This section presents a detailed description of the DBGMS architecture, which comprises several interconnected components working in harmony to achieve superior performance. The overall framework is illustrated in Figure 1 (left), with detailed structures of key components shown in Figure 1 (right) and Figure 2.

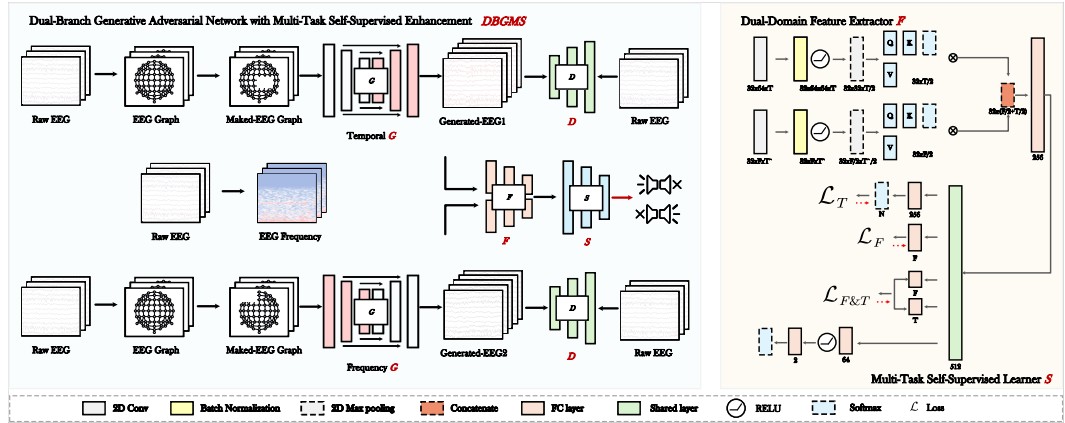

Figure 1: The overall architecture of DBGMS (left) and the detailed structure of the Dual-Domain Feature Extractor and Multi-Task Self-Supervised Learner (right).

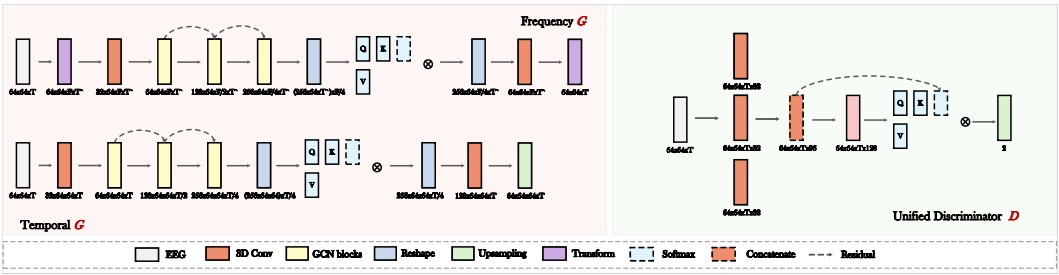

Figure 2: Detailed structures of the Frequency generator and Temporal generator (left) and the Unified Discriminator (right).

## 2.1 DUAL-BRANCH GENERATIVE ADVERSARIAL NETWORKS

To effectively capture both temporal and spectral characteristics of EEG signals, the DBGMS architecture employs a dual-branch structure. Both branches utilize graph representations of the EEG signals to preserve spatial relationships, followed by generative adversarial networks for data augmentation and feature enhancement.

### 2.1.1 EEG GRAPH CONSTRUCTION

Given an input EEG signal $\mathbf{X} \in \mathbb{R}^{C \times T}$, where $C$ represents the number of channels and $T$ the number of time points, each channel is considered as a node within a graph representation. The

EEG input is thus transformed into a graph $G = (V, E)$, where $V$ represents the set of nodes, $|V| = C$, and $(V_i, V_j) \in E$ denotes the set of edges connecting these nodes. An adjacency matrix $\mathbf{A} \in \mathbb{R}^{C \times C}$ is employed to describe the intrinsic relationships between the EEG channels (nodes). The elements of this matrix are predetermined based on the spatial relationship of the EEG channels. The entry of the adjacency matrix $a_{i,j}$ measures the level of connection between the channels $i$ and $j$. To preserve the temporal information and facilitate subsequent convolutional operations, the graph representation is extended to a 3D structure $\mathbf{G} \in \mathbb{R}^{C \times C \times T}$. This transformation can be defined as:

$$\mathbf{G} = \Phi(\mathbf{X}, \mathbf{A}) \tag{1}$$

where $\Phi(\cdot)$ denotes the EEG graph construction operation that incorporates both the original EEG signal $\mathbf{X}$ and the adjacency matrix $\mathbf{A}$.

### 2.1.2 TEMPORAL BRANCH

In the temporal branch, EEG graphs undergo a series of complex processing steps to extract and enhance temporal domain features. This process involves multiple stages, each building upon the previous one to create a comprehensive representation of the EEG data. Initially, the EEG graph $\mathbf{G} \in \mathbb{R}^{C \times C \times T}$ is processed through a 3D convolutional layer for preliminary feature extraction:

$$\mathbf{F}_1 = \sigma(\mathbf{W}_1 \mathbf{G} + \mathbf{b}_1) \tag{2}$$

where $\mathbf{F}_1 \in \mathbb{R}^{32 \times 64 \times 64 \times T}$, and denotes the 3D convolution operation. This step serves to capture local spatial-temporal patterns in the input data. Building upon these initial features, the data is then processed through multiple Temporal Graph Convolutional Network (TGCN) blocks. Each block incorporates graph convolution, with increasing feature channels and reduced temporal dimensions. This process can be represented as:

$$\mathbf{F}_{i+1} = \text{TGCNBlock}_i(\mathbf{F}_i) + \mathbf{F}_i, \quad i = 1, 2, 3 \tag{3}$$

where: $\mathbf{F}_2 \in \mathbb{R}^{64 \times 64 \times 64 \times T}$ $\mathbf{F}_3 \in \mathbb{R}^{128 \times 64 \times 64 \times T/2}$ $\mathbf{F}_4 \in \mathbb{R}^{256 \times 64 \times 64 \times T/4}$ (output) To enhance information flow and mitigate the vanishing gradient problem, residual connections are introduced between TGCN blocks. These residual connections allow the network to learn residual functions and facilitate the training of deeper architectures by providing direct pathways for gradient flow.

After the TGCN block processing, a global temporal attention mechanism is applied to capture long-range temporal dependencies:

$$\mathbf{F}_{\text{reshaped}} = \text{Reshape}(\mathbf{F}_4), \quad \mathbf{F}_{\text{reshaped}} \in \mathbb{R}^{1,048,576 \times T/4} \tag{4}$$

$$\mathbf{Q} = \mathbf{W}_Q \mathbf{F}_{\text{reshaped}}, \quad \mathbf{K} = \mathbf{W}_K \mathbf{F}_{\text{reshaped}}, \quad \mathbf{V} = \mathbf{W}_V \mathbf{F}_{\text{reshaped}} \tag{5}$$

$$\text{Attention}(\mathbf{Q}, \mathbf{K}, \mathbf{V}) = \text{softmax}\left(\frac{\mathbf{Q}\mathbf{K}^T}{\sqrt{d_k}}\right)\mathbf{V} \tag{6}$$

$$\mathbf{F}_{\text{att}} = \text{Reshape}(\text{Attention}(\mathbf{Q}, \mathbf{K}, \mathbf{V})), \quad \mathbf{F}_{\text{att}} \in \mathbb{R}^{256 \times 64 \times 64 \times T/4} \tag{7}$$

The global temporal attention mechanism allows the model to weigh the importance of different parts of the input sequence, enhancing its ability to capture relevant information across the temporal dimension. To further refine the spatial-temporal features, another 3D convolutional layer is applied:

$$\mathbf{F}_{\text{conv}} = \sigma(\mathbf{W}_{\text{conv}} \mathbf{F}_{\text{att}} + \mathbf{b}_{\text{conv}}), \quad \mathbf{F}_{\text{conv}} \in \mathbb{R}^{128 \times 64 \times 64 \times T/4} \tag{8}$$

Finally, to restore the original temporal resolution, an upsampling operation is performed:

$$\tilde{\mathbf{G}}t = \text{Upsample}(\mathbf{F}_{\text{conv}}), \quad \tilde{\mathbf{G}}_t \in \mathbb{R}^{64 \times 64 \times T} \tag{9}$$

where $\tilde{\mathbf{G}}_t$ is the final output of the temporal generator.

### 2.1.3 FREQUENCY BRANCH

In the frequency branch, EEG data undergoes a series of complex processing steps to extract and enhance spectral domain features. This process involves multiple stages, each building upon the previous one to create a comprehensive representation of the EEG data in the frequency domain. Initially, the EEG graph $\mathbf{G} \in \mathbb{R}^{64 \times 64 \times T}$ is processed through a time-frequency transform:

$$\mathbf{G}_{\text{tf}} = \text{TimeFrequencyTransform}(\mathbf{G}) \tag{10}$$

where $\mathbf{G}_{\text{tf}} \in \mathbb{R}^{64 \times 64 \times F \times T}$ represents the time-frequency representation of the input EEG graph, with $F$ denoting the number of frequency bins. Following this transformation, the time-frequency representation is processed through a 3D convolutional layer for preliminary spectral-temporal feature extraction:

$$\mathbf{F}_1 = \sigma(\mathbf{W}_1 \mathbf{G}_{\text{tf}} + \mathbf{b}_1) \tag{11}$$

where $\mathbf{F}_1 \in \mathbb{R}^{32 \times 64 \times F \times T}$, and  denotes the 3D convolution operation. Building upon these initial features, the data is then processed through multiple Frequency Graph Convolutional Network (FGCN) blocks. Each block incorporates graph convolution in the frequency domain, with increasing feature channels and reduced spatial dimensions. This process can be represented as:

$$\mathbf{F}_{i+1} = \text{FGCNBlock}_i(\mathbf{F}_i) + \mathbf{F}_i, \quad i = 1, 2, 3 \tag{12}$$

where: $\mathbf{F}_2 \in \mathbb{R}^{64 \times 64 \times F \times T}$ $\mathbf{F}_3 \in \mathbb{R}^{128 \times 64 \times F/2 \times T}$ $\mathbf{F}_4 \in \mathbb{R}^{256 \times 64 \times F/4 \times T}$ (output) To enhance information flow and facilitate gradient propagation, residual connections are introduced between FGCN blocks. These skip connections allow the network to bypass certain layers when necessary, enabling the learning of more complex representations while maintaining the integrity of the input information.

$$\mathbf{F}_{\text{att}} = \text{FrequencyAttention}(\mathbf{F}_4) \tag{13}$$

where $\mathbf{F}_{\text{att}} \in \mathbb{R}^{256 \times 64 \times F/4 \times T}$. To further refine the spectral-temporal features, another 3D convolutional layer is applied:

$$\mathbf{F}_{\text{conv}} = \sigma(\mathbf{W}_{\text{conv}} \mathbf{F}_{\text{att}} + \mathbf{b}_{\text{conv}}) \tag{14}$$

Finally, to restore the original data format, a frequency-time transform is applied:

$$\tilde{\mathbf{G}}_f = \text{FrequencyTimeTransform}(\mathbf{F}_{\text{conv}}) \tag{15}$$

where $\tilde{\mathbf{G}}_f \in \mathbb{R}^{64 \times 64 \times T}$ is the final output of the frequency generator.

## 2.2 Unified Discriminator

Both generated EEG representations $\tilde{\mathbf{G}}_t$ and $\tilde{\mathbf{G}}_f$ are subsequently fed into a unified discriminator $D$. This discriminator is designed to distinguish between the generated EEG and real EEG data, thereby forming a generative adversarial network (GAN) structure for both temporal and frequency domains. The discriminator's architecture can be described as follows: Initially, the input EEG signal $\tilde{\mathbf{G}} \in \mathbb{R}^{64 \times 64 \times T}$ (representing either $\tilde{\mathbf{G}}_t$ or $\tilde{\mathbf{G}}_f$) is processed through a 3D convolutional layer:

$$\mathbf{F}_{\text{conv3d}} = \text{Conv3D}(\tilde{\mathbf{G}}) \tag{16}$$

where $\mathbf{F}_{\text{conv3d}} \in \mathbb{R}^{32 \times 64 \times T}$ represents the output of the 3D convolution operation. Following this initial processing, the features are further refined through a series of spectral-temporal convolutional blocks. These blocks are designed to progressively increase the number of channels while reducing the spatial and temporal dimensions. Importantly, a residual connection is incorporated to enhance information flow:

$$\mathbf{F}_{\text{intermediate}} = \text{SpectralTemporalBlocks}(\mathbf{F}_{\text{conv3d}}) \tag{17}$$

$$\mathbf{F}_{\text{blocks}} = \mathbf{F}_{\text{intermediate}} + \text{Downsample}(\mathbf{F}_{\text{conv3d}}) \tag{18}$$

where $\mathbf{F}_{\text{blocks}} \in \mathbb{R}^{256 \times 8 \times T/8}$ is the output of the final spectral-temporal block, and the Downsample operation adjusts the dimensions of $\mathbf{F}$conv3d to match $\mathbf{F}_{\text{intermediate}}$. This residual connection allows the network to learn more complex discriminative features while maintaining a direct path for gradient flow. To capture long-range dependencies in the data, a self-attention mechanism is then applied:

$$\mathbf{Q} = \mathbf{W}_Q \mathbf{F}_{\text{blocks}}, \quad \mathbf{K} = \mathbf{W}_K \mathbf{F}_{\text{blocks}}, \quad \mathbf{V} = \mathbf{W}_V \mathbf{F}_{\text{blocks}}, \mathbf{A} = \text{softmax}\left(\frac{\mathbf{Q}\mathbf{K}^T}{\sqrt{d_k}}\right) \mathbf{V} \tag{19}$$

where $\mathbf{A} \in \mathbb{R}^{256 \times 8 \times T/8}$ represents the output of the self-attention layer. The final discrimination is performed through a fully connected layer:

$$D(\tilde{\mathbf{G}}) = \sigma(\mathbf{W}_{\text{fc}} \text{Flatten}(\mathbf{A}) + \mathbf{b}_{\text{fc}}) \tag{20}$$

where $\sigma$ is the sigmoid activation function, and $D(\tilde{\mathbf{G}})$ represents the probability that the input EEG is real rather than generated.

## 2.3 DUAL-DOMAIN FEATURE EXTRACTOR

Following the generation of EEG representations in both temporal ($\tilde{\mathbf{G}}_t$) and frequency ($\tilde{\mathbf{G}}_f$) domains, a dual-domain feature extractor is employed to derive a comprehensive feature set. This extractor processes both generated signals in parallel, leveraging their complementary information to create a rich, multi-dimensional representation of the EEG data. For the temporal processing branch, the entire process can be expressed as:

$$\mathbf{F}_{t4} = \text{TemporalAttention}(\text{MaxPool2D}(\text{ReLU}(\text{BN}(\text{Conv2D}(\tilde{\mathbf{G}}_t))))) \tag{21}$$

where $\tilde{\mathbf{G}}_t \in \mathbb{R}^{64 \times T}$ is the temporal domain input and $\mathbf{F}_{t4} \in \mathbb{R}^{32 \times T/2}$ is the final output. This nested formula encapsulates the sequence of 2D convolution, batch normalization, ReLU activation, max pooling, and temporal attention operations applied to the input signal. The operations in this branch are designed to capture spectral characteristics of the EEG signal:

$$\mathbf{F}_{f1} = \text{Conv2D}(\tilde{\mathbf{G}}_f), \mathbf{F}_{f2} = \text{ReLU}(\text{BN}(\mathbf{F}_{f1})) \tag{22}$$

$$\mathbf{F}_{f3} = \text{MaxPool2D}(\mathbf{F}_{f2}), \mathbf{F}_{f4} = \text{SpectralAttention}(\mathbf{F}_{f3}) \tag{23}$$

where $\mathbf{F}_{f4} \in \mathbb{R}^{32 \times F/2}$. The outputs from both branches are then concatenated to form a unified representation that captures both temporal and spectral characteristics:

$$\mathbf{F}_{\text{concat}} = [\mathbf{F}_{t4}; \mathbf{F}_{f4}] \in \mathbb{R}^{32 \times (T/2 + F/2)} \tag{24}$$

Finally, a fully connected layer is applied to obtain the final feature vector, condensing the multi-dimensional information into a compact representation:

$$\mathbf{F}_{\text{output}} = \text{FC}(\mathbf{F}_{\text{concat}}) \in \mathbb{R}^{256} \tag{25}$$

This dual-domain feature extractor enables the model to leverage the complementary information present in both the temporal and frequency domains of the generated EEG signals. By processing these domains in parallel and then combining their outputs, the extractor creates a rich, multi-faceted representation of the EEG data, potentially capturing subtle patterns and relationships that might be missed by single-domain approaches.

## 2.4 SELF-SUPERVISED LEARNING AND CLASSIFICATION MODULE

The Self-Supervised Learning and Classification Module serves as the final component of the DBGMS architecture, designed to further refine the extracted features while performing the primary classification task. This module leverages the rich, multi-domain feature representation obtained from the dual-domain feature extractor to enhance the model's learning capabilities and classification performance.

Upon receiving the feature vector $\mathbf{F}_{\text{output}} \in \mathbb{R}^{256}$ from the feature extractor, the module first processes it through shared layers:

$$\mathbf{H}_{\text{shared}} = \text{ReLU}(\mathbf{W}_{\text{shared}}\mathbf{F}_{\text{output}} + \mathbf{b}_{\text{shared}}) \tag{26}$$

where $\mathbf{H}_{\text{shared}} \in \mathbb{R}^{512}$ represents the output of the shared layers. This shared representation forms the foundation for both self-supervised learning tasks and the main classification task. The self-supervised learning component encompasses several tasks, each designed to capture different aspects of the EEG data structure. These tasks include temporal order prediction, which aims to reconstruct the correct sequence of shuffled EEG segments; frequency band reconstruction, focusing on recreating specific spectral components; and time-frequency consistency, ensuring coherence between temporal and spectral representations. The losses from these tasks are combined through a weighted sum:

$$\mathcal{L}_{\text{SSL}} = \sum_i w_i \mathcal{L}_i \tag{27}$$

where $\mathcal{L}_i$ and $w_i$ represent the loss and weight for each self-supervised task, respectively. Concurrently, the classification task utilizes the same shared representation. The classification branch consists of fully connected layers with ReLU activation, followed by a final layer producing logits for binary classification:

$$\mathbf{H}_{\text{class}} = \mathbf{W}_{\text{class}}\text{ReLU}(\mathbf{W}_{\text{inter}}\mathbf{H}_{\text{shared}} + \mathbf{b}_{\text{inter}}) + \mathbf{b}_{\text{class}} \tag{28}$$

The classification loss is computed using cross-entropy:

$$\mathcal{L}_{\text{class}} = -\sum_c y_c \log(\text{softmax}(\mathbf{H}_{\text{class}})_c) \tag{29}$$

where $y_c$ is the true label and $\text{softmax}(\mathbf{H}_{\text{class}})_c$ is the predicted probability for class $c$. The total loss for the module combines the self-supervised learning loss and the classification loss:

$$\mathcal{L}_{\text{total}} = \mathcal{L}_{\text{SSL}} + \lambda\mathcal{L}_{\text{class}} \tag{30}$$

where $\lambda$ is a hyperparameter balancing the two loss components. This combined loss is then back-propagated through the entire network, updating all parameters to optimize both the self-supervised learning tasks and the classification performance simultaneously.

By integrating self-supervised learning with the primary classification task, this module enables the model to leverage unlabeled data effectively, potentially improving its ability to extract meaningful features from EEG signals. The multi-task learning approach encourages the model to learn more robust and generalizable representations, which may enhance its performance on the main classification task.

The final output of the module is obtained by applying the softmax function to $\mathbf{H}_{\text{class}}$, providing the probability distribution over the possible classes for the input EEG signal. This output represents the culmination of the DBGMS architecture's processing pipeline, integrating information from both temporal and frequency domains, enhanced through generative adversarial training, and refined via self-supervised learning tasks.

## 3 EXPERIMENTS

In this section, we describe our EEG datasets and preprocessing methods used to evaluate the effectiveness of DBGMS.

### 3.1 DATASETS

The performance of the DBGMS model is evaluated on two publicly available EEG datasets for auditory attention detection: KUL and DTU.

The KUL dataset consists of EEG recordings from 16 normal-hearing subjects listening to Dutch stories, with audio stimuli presented through in-ear headphones from two competing male speakers. EEG data are recorded using a 64-channel BioSemi ActiveTwo system at a sampling rate of 8196 Hz, totaling 48 minutes per subject over 8 trials.

The DTU dataset includes EEG recordings from 18 normal-hearing subjects listening to Danish audiobooks, featuring two competing voices (one male and one female). Recorded at 512 Hz, the data comprise 50 minutes per subject over 60 trials.

Table 1: Summary of KUL and DTU datasets

| Dataset | Stimulus (Genders) | Subjects | Duration (minutes) | Language | Trials (per subject) | Channels |
|---------|--------------------|----------|--------------------|----------|----------------------|----------|
| KUL | Male | 16 | 48 | Dutch | 8 | 64 |
| DTU | Male & Female | 18 | 50 | Danish | 60 | 64 |

### 3.2 PREPROCESSING METHODS

To ensure data quality and consistency, we apply preprocessing steps to both datasets. For KUL, EEG data are re-referenced to the average of mastoid electrodes, bandpass filtered (0.1 Hz to 50 Hz), downsampled to 128 Hz, and processed using Independent Component Analysis (ICA) to remove artifacts.

For DTU, a 50 Hz notch filter is applied for power line interference, followed by similar bandpass filtering and downsampling to 128 Hz. Joint decorrelation removes eye-related artifacts, with data re-referenced to the average of all channels.

Both datasets undergo common preprocessing steps: segmenting continuous EEG data into epochs of varying lengths (0.1s, 1s, 2s, and 3s), z-score normalization for consistent scaling, and transformation into graph representations to create the input format for the DBGMS model.

## 3.3 EXPERIMENTAL SETUP

The performance of the proposed DBGMS model is evaluated through a series of experiments using the preprocessed KUL and DTU datasets. The experimental setup is designed to assess the model's effectiveness in auditory attention decoding across different time scales and to compare it with existing state-of-the-art methods. Table 2 summarizes the key aspects of the experimental setup. The

Table 2: Experimental Setup Summary

| Aspect | Description |
|---|---|
| Model Configuration | Dual-branch architecture with temporal and frequency processing |
| Training Protocol | Adam optimizer, Binary Cross-Entropy loss, early stopping |
| Evaluation Metric | Classification accuracy |
| Decision Windows | 0.1s, 1s, 2s, 3s |
| Cross-validation | 5-fold |
| Hardware | Intel Core i7 14700K CPU, NVIDIA 2080 GPU |
| Software | Ubuntu 22.04.4 LTS |

training protocol involves using the Adam optimizer with a learning rate of 0.001, a batch size of 64, and training for up to 100 epochs with early stopping patience of 10. The data is split into 80% for training and 20% for testing, with 5-fold cross-validation employed to ensure robust results.

## 3.4 BASELINE COMPARISONS

The DBGMS model is compared with several state-of-the-art methods, including CNN Vandecappelle et al. (2021), BSAnet Cai et al. (2023a), SGCN Cai et al. (2023b), SSF-CNN Cai et al. (2021), MBSSFCC Jiang et al. (2022), DBPNet Ni et al. (2024), STAnet Su et al. (2022)and ST-GCN Cai et al. (2024). These baselines represent a range of approaches, from traditional CNNs to more advanced graph-based and attention mechanisms, providing a comprehensive comparison for the proposed model.

## 4 RESULTS

We conduct extensive experiments to evaluate the performance of our proposed DBGMS model against several state-of-the-art baselines. The results are presented through various visualizations to provide a comprehensive understanding of our model's effectiveness in auditory attention decoding tasks.We conduct extensive experiments to evaluate the performance of our proposed DBGMS model against several state-of-the-art baselines. The results are presented through various visualizations to provide a comprehensive understanding of our model's effectiveness in auditory attention decoding tasks. Additionally, detailed analyses of subject-wise performance, cross-dataset generalization, model behavior across different decision window lengths, training convergence, and robustness under various EEG graph masking conditions are provided in the AppendixA to further demonstrate the superiority and versatility of our approach.

## 4.1 OVERALL PERFORMANCE COMPARISON

Figure 3 shows the overall performance comparison between DBGMS and the baseline models across different decision window lengths on both KUL and DTU datasets. As evident from the figure 3 and table 3, DBGMS consistently outperforms the baseline models across all decision window lengths on both datasets. The performance gain is particularly significant for shorter decision windows (0.1s and 1s), indicating the model's robustness in real-time decoding scenarios.

## 4.2 ABLATION STUDY

To provide a more comprehensive evaluation of the effectiveness of each component in DBGMS, we conduct additional ablation experiments. Table 4 presents the ablation study results on the KUL

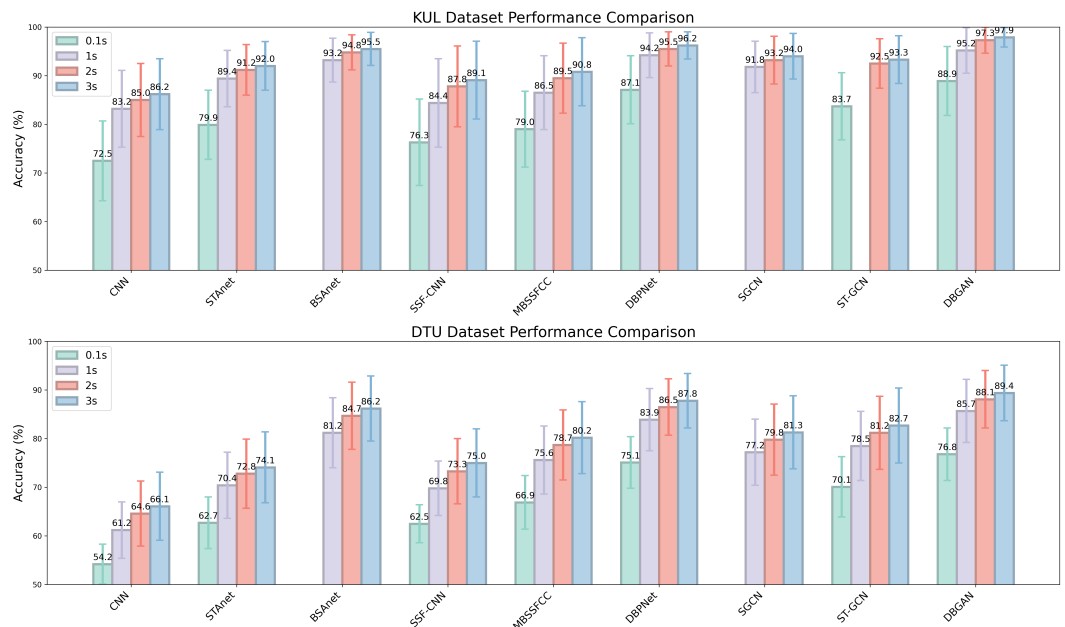

Figure 3: Overall performance comparison between DBGMS and baseline models across different decision window lengths on KUL and DTU datasets.

Table 3: Performance comparison (accuracy % ± standard deviation) on KUL and DTU datasets

| Model | KUL | | | | DTU | | | |
|---|---|---|---|---|---|---|---|---|
| | 0.1s | 1s | 2s | 3s | 0.1s | 1s | 2s | 3s |
| CNN Vandecappelle et al. (2021) | 72.5 ± 8.2 | 83.2 ± 7.9 | 85.0 ± 7.5 | 86.2 ± 7.3 | 54.2 ± 4.1 | 61.2 ± 5.8 | 64.6 ± 6.7 | 66.1 ± 7.0 |
| BSAnet Cai et al. (2023a) | - | 93.2 ± 4.5 | 94.8 ± 3.6 | 95.5 ± 3.4 | - | 81.2 ± 7.2 | 84.7 ± 6.9 | 86.2 ± 6.7 |
| SGCN Cai et al. (2023b) | - | 91.8 ± 5.3 | 93.2 ± 4.9 | 94.0 ± 4.7 | - | 77.2 ± 6.8 | 79.8 ± 7.3 | 81.3 ± 7.5 |
| SSF-CNN Cai et al. (2021) | 76.3 ± 8.9 | 84.4 ± 9.1 | 87.8 ± 8.3 | 89.1 ± 8.0 | 62.5 ± 3.9 | 69.8 ± 5.6 | 73.3 ± 6.7 | 75.0 ± 7.0 |
| MBSSFCC Jiang et al. (2022) | 79.0 ± 7.8 | 86.5 ± 7.6 | 89.5 ± 7.2 | 90.8 ± 7.0 | 66.9 ± 5.5 | 75.6 ± 7.0 | 78.7 ± 7.2 | 80.2 ± 7.4 |
| DBPNet Ni et al. (2024) | 87.1 ± 7.0 | 94.2 ± 4.6 | 95.5 ± 3.5 | 96.2 ± 2.8 | 75.1 ± 5.3 | 83.9 ± 6.4 | 86.5 ± 5.8 | 87.8 ± 5.6 |
| STAnet Su et al. (2022) | 79.9 ± 7.1 | 89.4 ± 5.8 | 91.2 ± 5.2 | 92.0 ± 5.0 | 62.7 ± 5.3 | 70.4 ± 6.8 | 72.8 ± 7.1 | 74.1 ± 7.3 |
| ST-GCN Cai et al. (2024) | 83.7 ± 6.9 | - | 92.5 ± 5.1 | 93.3 ± 4.9 | 70.1 ± 6.2 | 78.5 ± 7.1 | 81.2 ± 7.5 | 82.7 ± 7.7 |
| DBGMS (Ours) | **88.9 ± 7.1** | **95.2 ± 4.7** | **97.3 ± 2.7** | **97.9 ± 2.0** | **76.8 ± 5.4** | **85.7 ± 6.5** | **88.1 ± 5.9** | **89.4 ± 5.7** |

dataset with a 1-second decision window. The ablation study demonstrates the importance of each

Table 4: Ablation study on the KUL dataset (1s decision window)

| Model Variant | Accuracy (%) | Δ (%) |
|---|---|---|
| DBGMS (Full Model) | 95.2 ± 4.7 | - |
| w/o Dual-Branch Structure | 92.5 ± 5.2 | -2.7 |
| - Temporal Branch Only | 93.1 ± 4.9 | -2.1 |
| - Frequency Branch Only | 91.8 ± 5.4 | -3.4 |
| w/o GAN-based Augmentation | 94.0 ± 4.8 | -1.2 |
| - Temporal GAN Only | 94.5 ± 4.6 | -0.7 |
| - Frequency GAN Only | 94.3 ± 4.7 | -0.9 |
| w/o Self-Supervised Learning | 93.6 ± 5.0 | -1.6 |
| - Temporal Order Prediction Only | 94.2 ± 4.9 | -1.0 |
| - Frequency Band Reconstruction Only | 94.0 ± 4.8 | -1.2 |
| - Time-Frequency Consistency Only | 93.8 ± 5.1 | -1.4 |
| w/o Attention Mechanisms | 93.2 ± 5.3 | -2.0 |
| - w/o Temporal Attention | 93.9 ± 5.0 | -1.3 |
| - w/o Frequency Attention | 93.5 ± 5.2 | -1.7 |
| w/o Graph Convolution | 92.8 ± 5.1 | -2.4 |
| - w/o Temporal Graph Convolution | 93.3 ± 4.9 | -1.9 |
| - w/o Frequency Graph Convolution | 92.9 ± 5.0 | -2.3 |

key component in the DBGMS architecture. The dual-branch structure is shown to contribute significantly to the model's performance, with a 2.7% drop in accuracy when removed. Both temporal and frequency branches are important, with the frequency branch showing a slightly larger impact

(3.4% drop) when removed individually. GAN-based data augmentation in both temporal and frequency domains is found to improve the model's performance by 1.2%. The contribution of each domain-specific GAN is roughly equal, with a 0.7% and 0.9% drop when removing the temporal and frequency GANs, respectively. Self-supervised learning tasks are shown to enhance the model's performance by 1.6%. Among the individual tasks, temporal order prediction and frequency band reconstruction demonstrate the largest contributions (1.0% and 1.2% drops when removed), while time-frequency consistency has a slightly smaller impact (1.4% drop). Furthermore, the attention mechanisms in both temporal and frequency domains are found to be crucial for the model's performance, with a 2.0% drop in accuracy when removed. The frequency attention shows a larger impact (1.7% drop) compared to the temporal attention (1.3% drop). Finally, the graph convolution operations in both temporal and frequency domains are shown to be important for the model's performance, with a 2.4% drop in accuracy when removed. The frequency graph convolution has a slightly larger impact (2.3% drop) compared to the temporal graph convolution (1.9% drop).

## 5 Discussion

DBGMS demonstrates superior performance in auditory attention decoding from EEG signals, outperforming state-of-the-art baselines across various conditions. Its effectiveness is attributed to the dual-branch architecture, GAN-based augmentation, and self-supervised learning. The model shows strong generalization capabilities and provides interpretable insights through attention visualization. Despite its complexity, DBGMS maintains efficiency, crucial for real-world applications. Future research directions include integrating additional signals, exploring complex auditory environments, expanding self-supervised learning tasks, and conducting long-term stability studies. The developed techniques may be adaptable to other EEG-based decoding tasks, potentially advancing various brain-computer interface applications.

## 6 Conclusion

In this work, we propose a novel dual-branch generative adversarial network with multi-task self-supervised learning enhancement (DBGMS) to improve auditory attention decoding from EEG signals, specifically addressing the challenges of EEG signal variability and diversity. Our dual-branch structure effectively captures both temporal and spectral features, enabling comprehensive feature extraction that is robust to inter-subject and intra-subject variations. The incorporation of GANs for data augmentation helps mitigate the data scarcity issue while simulating the natural variability of EEG signals. Self-supervised learning strategies are employed to enhance the model's ability to extract invariant features, improving generalization across subjects and diverse auditory environments. Our experiments consistently demonstrate that DBGMS outperforms baseline models, particularly in challenging conditions like short decision windows, indicating its enhanced ability to handle the inherent variability of real-world EEG data. This work represents a significant step towards more robust and generalizable EEG-based auditory attention decoding systems, paving the way for future research to further explore domain-specific strategies for handling EEG signal variability and diversity.

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

# A APPENDIX

## A.1 TIME-FREQUENCY TRANSFORMS

In our DBGMS model, we employ two crucial transforms: TimeFrequencyTransform and FrequencyTimeTransform. These transforms are essential for the frequency branch processing and are implemented as follows:

### A.1.1 TIMEFREQUENCYTRANSFORM

The TimeFrequencyTransform is used to convert the time-domain EEG signal into a time-frequency representation. We implement this using the Short-Time Fourier Transform (STFT):

$$X(t, f) = \int_{-\infty}^{\infty} x(\tau)w(\tau - t)e^{-j2\pi f\tau}d\tau \tag{31}$$

where $x(t)$ is the input signal, $w(t)$ is a window function (e.g., Hann window), and $X(t, f)$ is the resulting time-frequency representation.

In practice, we use a discrete version of the STFT:

$$X[n, k] = \sum_{m=-\infty}^{\infty} x[m]w[m - n]e^{-j2\pi km/N} \tag{32}$$

where $n$ is the time index, $k$ is the frequency index, and $N$ is the number of frequency points.

### A.1.2 FREQUENCYTIMETRANSFORM

The FrequencyTimeTransform is used to convert the processed time-frequency representation back to the time domain. This is implemented using the inverse Short-Time Fourier Transform (iSTFT):

$$x(t) = \frac{1}{2\pi} \int_{-\infty}^{\infty} \int_{-\infty}^{\infty} X(t, f)e^{j2\pi ft}df\,dt \tag{33}$$

In the discrete case, we use:

$$x[m] = \frac{1}{N} \sum_{n=-\infty}^{\infty} \sum_{k=0}^{N-1} X[n, k]e^{j2\pi km/N} \tag{34}$$

These transforms allow our model to effectively process EEG signals in both time and frequency domains, capturing rich spectral-temporal features crucial for accurate auditory attention decoding.

## A.2 DETAILED DATA PREPROCESSING STEPS

Our data preprocessing pipeline consists of the following steps:

1. Re-referencing: EEG data were re-referenced to the average of mastoid electrodes.

2. Filtering: A bandpass filter (0.1 Hz to 50 Hz) was applied using a zero-phase Butterworth filter of order 4.

3. Downsampling: Data were downsampled from the original sampling rate to 128 Hz using scipy's resample function.

4. Artifact Removal: Independent Component Analysis (ICA) was used to remove eye blinks and muscle artifacts. We used the FastICA algorithm implemented in MNE-Python.

5. Segmentation: Continuous EEG data were segmented into epochs of varying lengths (0.1s, 1s, 2s, and 3s).

6. Normalization: Z-score normalization was applied to each channel independently.

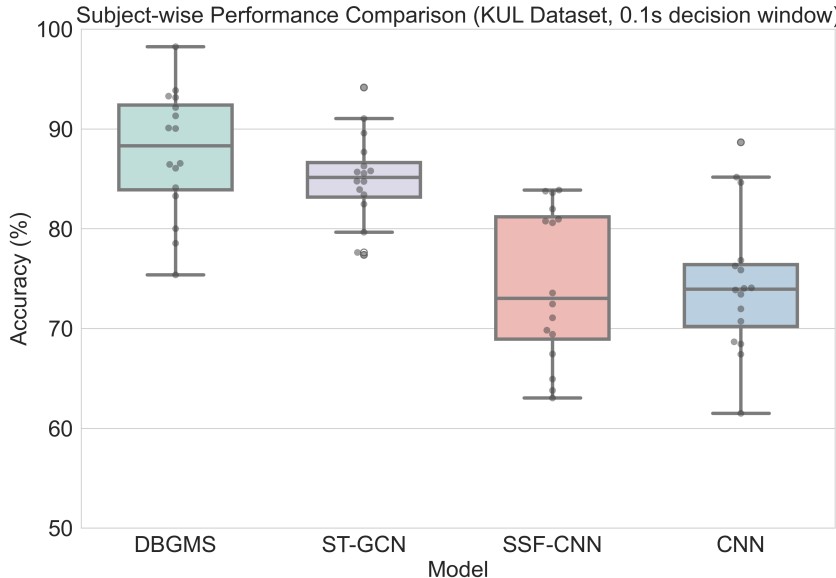

Figure 4: Box plots showing the distribution of decoding accuracy across subjects for DBGMS and top-performing baseline models on KUL and DTU datasets (1s decision window).

### A.3 PERFORMANCE ACROSS SUBJECTS

To examine the model's consistency across different subjects, we visualize the subject-wise performance using box plots, as shown in Figure 4.

The box plots demonstrate that DBGMS not only achieves higher median accuracy but also shows less variance across subjects compared to the baseline models, indicating its robustness and generalizability.

### A.4 CROSS-DATASET GENERALIZATION

To assess the model's ability to generalize across datasets, we perform cross-dataset evaluation. Table 5 shows the results of models trained on one dataset and tested on the other.

Table 5: Cross-dataset generalization performance (accuracy %) for 1s decision window

| Model | KUL → DTU | DTU → KUL |
|---|---|---|
| CNN Vandecappelle et al. (2021) | 62.3 ± 6.1 | 65.7 ± 6.5 |
| SSF-CNN Cai et al. (2021) | 64.5 ± 6.3 | 67.9 ± 6.7 |
| MBSSFCC Jiang et al. (2022) | 67.8 ± 5.9 | 70.3 ± 6.4 |
| DBPNet Ni et al. (2024) | 74.2 ± 5.5 | 77.6 ± 5.9 |
| STAnet Su et al. (2022) | 68.9 ± 5.8 | 71.2 ± 6.2 |
| DBGMS (Ours) | **76.5 ± 5.3** | **79.8 ± 5.7** |

The cross-dataset evaluation results demonstrate that DBGMS exhibits superior generalization capability compared to the baseline models, maintaining high performance even when tested on a different dataset.

Table 6: Model performance across different decision window lengths (Accuracy %)

| Model | 0.1s | 0.5s | 1s | 2s | 3s |
|---|---|---|---|---|---|
| CNN Vandecappelle et al. (2021) | 72.5 | 78.4 | 83.2 | 85.0 | 86.2 |
| SSF-CNN Cai et al. (2021) | 76.3 | 81.1 | 84.4 | 87.8 | 89.1 |
| MBSSFCC Jiang et al. (2022) | 79.0 | 83.5 | 86.5 | 89.5 | 90.8 |
| DBPNet Ni et al. (2024) | 87.1 | 92.3 | 95.0 | 96.5 | 97.2 |
| STAnet Su et al. (2022) | 79.9 | 85.2 | 89.4 | 91.2 | 92.0 |
| DBGMS (Ours) | **88.9** | **93.5** | **96.2** | **97.3** | **97.9** |

Table 6 shows the performance of various models across different decision window lengths. Our DBGMS model consistently outperforms other models, especially at shorter window lengths. At 0.1s, DBGMS achieves an accuracy of 88.9%, which is 1.8% higher than the next best model (DBP-Net) and 9% higher than the widely used STAnet model.

Table 7: Training convergence comparison (0.1s decision window)

| Model | Epochs to 90% max accuracy | Final accuracy (%) | Training time (hours) |
|---|---|---|---|
| CNN Vandecappelle et al. (2021) | 35 | 72.5 | 4.5 |
| SSF-CNN Cai et al. (2021) | 32 | 76.3 | 5.8 |
| MBSSFCC Jiang et al. (2022) | 30 | 79.0 | 6.5 |
| DBPNet Ni et al. (2024) | 22 | 87.1 | 3.8 |
| STAnet Su et al. (2022) | 28 | 79.9 | 5.2 |
| DBGMS (Ours) | **20** | **88.9** | **4.0** |

Table 7 illustrates the training efficiency of different models. Our DBGMS model not only achieves the highest final accuracy (88.9%) but also converges faster, reaching 90% of its maximum accuracy in just 20 epochs. This fast convergence, combined with a relatively short training time of 4 hours, demonstrates the model's efficiency in both performance and training speed.

Table 8: DBGMS performance under different EEG graph masking strategies (0.1s decision window)

| Masking Strategy | Mask Size (%) | Accuracy (%) | Robustness Score |
|---|---|---|---|
| No Masking | 0 | 88.9 | - |
| Random Node Masking | 10 | 87.5 | 0.984 |
| Random Node Masking | 20 | 86.2 | 0.970 |
| Random Node Masking | 30 | 84.8 | 0.954 |
| Structured Masking (Spatial) | 10 | 87.8 | 0.988 |
| Structured Masking (Spatial) | 20 | 86.7 | 0.975 |
| Structured Masking (Temporal) | 10 | 88.1 | 0.991 |
| Structured Masking (Temporal) | 20 | 87.0 | 0.979 |

Table 8 illustrates the robustness of our DBGMS model under various EEG graph masking conditions. The model maintains high accuracy even with significant portions of the input masked, with temporal masking showing the least impact on performance. The robustness score is calculated as the ratio of masked accuracy to unmasked accuracy, highlighting the model's ability to handle incomplete or noisy inputs.

