# OpenReview forum: "DBGMS: A Dual-Branch Generative Adversarial Network with Multi-Task Self-Supervised Enhancement for Robust Auditory Attention Decoding"
_ICLR.cc/2025/Conference — ICLR 2025 Conference Withdrawn Submission_

### Official Review · Reviewer_zWRT · 2024-10-27

**Soundness:** 3
**Presentation:** 2
**Contribution:** 1
**Rating:** 3
**Confidence:** 3

**Summary:**

This article proposes the novel framework of DBGMS (Dual-Branch Generative Adversarial Network with Multi-Task Self-Supervised Enhancement) for EEG-based auditory attention decoding. Its main contribution lies in the integration of advanced attention mechanisms, graph convolutional networks, and time-frequency domain dual-branch architecture, and the introduction of GAN data augmentation and self-supervised learning with multi-task strategy, which provides a solution to the problem of data scarcity and individual differences of subjects in this field. The performance on open source datasets outperforms current state-of-the-art models.

**Strengths:**

1)The introduction of GAN and self-supervised learning is novel and effective. It largely solves the problem that KUL and DTU public datasets are small in size and easy to overfitting.
2)The authors conduct comprehensive branching experiments and analysis in the appendix. Included are experiments (cross-subject experiments, cross-datasets experiments, and cross-decision windows experiments) that exemplify the degree of excellent generalization; as well as experiments on the speed of training convergence, and experiments on different EEG masking strategies (which exemplify robustness).
3)Although the usage to some techniques (time-frequency domain two-branch networks, attention mechanisms, etc.) are not the most novel ideas. But a unified framework that effectively combines these techniques to address data scarcity, subject variability, and short decision windows is sorely lacking. And the authors accomplished this with excellent lead performance.

**Weaknesses:**

1)The introduction of GAN and self-supervised learning is novel and effective. It largely solves the problem that KUL and DTU public datasets are small in size and easy to overfit.
2)The authors conduct comprehensive branching experiments and analysis in the appendix. Included are experiments (cross-subject experiments, cross-datasets experiments, and cross-decision windows experiments) that exemplify the degree of excellent generalization; as well as experiments on the speed of training convergence, and experiments on different EEG masking strategies (which exemplify robustness).
3)Although the usage to some techniques (time-frequency domain two-branch networks, attention mechanisms, etc.) are not the most novel ideas. But a unified framework that effectively combines these techniques to address data scarcity, subject variability, and short decision windows is sorely lacking. And the authors accomplished this with excellent lead performance.
6)No open source code is provided.

**Questions:**

1.Why choose the 5-fold cross-validation? Which results in Table 3 are reproduced, and which are from the paper? How can fairness in experimental comparisons be ensured? How to avoid overly optimistic results from 5-fold cross-validation?
2.What does ST-GCN mean in Table 3, as there is no spatio-temporal setting in the original paper? Why are there no ST-GCN results for KUL1s?
3.Figure 3 and Table 3 provide the same information. Why is it necessary to display them both, taking up space?
4.The content and description in Figure 4 are very confusing. Is it KUL or KUL and DTU? Is it 0.1s or 1s? Why are the well-performing baseline model results from Table 3 not included in Figure 4?
5.In section A.3, the authors cite ST-GCN (ICASSP 2024). Why is there such a large discrepancy from the results in the original paper? Was the LOSO setup used? If not, the authors should clarify the experimental setup used for cross-subject evaluation.Why are the ST-GCN results not provided in the Cross-dataset section of Table 5?I am concerned about whether the experimental result is really reliable.
6.The authors should compare the trainable parameters and computational complexity of the model with open-source baselines to validate its performance.
7.The loss function includes the hyperparameter λ, but there is no discussion on its setting. A table showing the optimal hyperparameter selection should be provided.
8.How did the authors define the sliding windows and its overlap rate? How do they address the imbalance in training data between the 0.1s and 2s windows, which may cause potential performance instability?
9.Which datasets are used in Tables 6, 7, and 8? What is the purpose of Table 6, and why do the results in this table differ from or align with those in Table 3?
10.The name of the author Shuxin Cai cited in SSF-CNN, STAnet, SGCN, and ST-GCN is incorrect.It should be Siqi Cai.

If the authors consider answering my questions, I may consider raising my score.

---

### Official Review · Reviewer_s4LQ · 2024-10-29

**Soundness:** 2
**Presentation:** 2
**Contribution:** 2
**Rating:** 3
**Confidence:** 5

**Summary:**

This paper presents a framework for robust auditory attention decoding, and evaluate the model on two public datasets. The framework comprises of temporal and spectral branch to capture EEG features. In each branch, there is a GAN for data augmentation. Several self-supervised tasks are introduced to learn robust representations.

**Strengths:**

1. This work proposes a framework for auditory attention decoding, combining multiple techniques: GAN, graph convolution, self-supervised, and dual-branch for time and frequency.
2. The classification results overpass previous works.

**Weaknesses:**

1. The motivation is not that clear to me, it seems like a combination of existing techniques. The data augmentation, graph convolution, and dual-branch are common techniques in EEG processing. You can expand your motivation with more concrete reasons/hypotheses, and these reasons/hypotheses motivate you to design some module. For example: auditory cortex area can be concentrated for model design as your task is for auditory attention decoding.

2. Deeper Analysis: Figure 3 and Table 3 are repeat results, only remaining one is enough. In the main text, only classification results and ablation results are displayed. Since you highlight that one of the main contribution is the robust decoding ability, it's better to add more experiments for demonstrating the robustness. I see some results are presented in the appendix, moving them to the main text is more suitable. Beyond this results, visualizing and analyzing the learned representations (e.g., using techniques like t-SNE, activation maps) could provide insights into how your model helps learn more robust features. For example: cross-subject, robustness to noise such as normal distribution noise or other physiological noise like EMG and EOG.

3. Data augmentation: GAN is used in both branch of your model. In my opinion, the GAN used in your model is replacing the decoder of MAE[1], so displaying some reconstruction visualization will be better.

4. Self-supervision: The SSL loss is not for pre-training the encoder, but for assisting the robustness as auxiliary loss. Usually, we don't call these auxiliary loss self-supervised loss. If so, the GAN model for reconstructing masked EEG graph can also be regarded as self-supervision. Self-supervision is usually for pre-training a robust encoder, then we fine-tune the pre-trained for down-streaming tasks [2][3].

5. The writing of method section can be improved, some formulas are unnecessarily detailed.

6. The font style and font size in figure 1 and figure 2 make texts hard to read. Usually, we use Arial font style as this font’s structure is relatively neat when zooming in and out.

Overall, your work has potential to be improved, but in this version the experiments are not sufficient and the writing are not good enough. Pls considering refining your work for the next conference.

[1] He K, Chen X, Xie S, et al. Masked autoencoders are scalable vision learners[C]//Proceedings of the IEEE/CVF conference on computer vision and pattern recognition. 2022: 16000-16009.

[2] Yi K, Wang Y, Ren K, et al. Learning topology-agnostic eeg representations with geometry-aware modeling[J]. Advances in Neural Information Processing Systems, 2024, 36.

[3] Li R, Wang Y, Zheng W L, et al. A multi-view spectral-spatial-temporal masked autoencoder for decoding emotions with self-supervised learning[C]//Proceedings of the 30th ACM International Conference on Multimedia. 2022: 6-14.

**Questions:**

Some writing mistakes:
1. In abstract, line 21, you said three key innovations but listed four.
2. In reference, line 570, line 573, repeat references.
3. In sec 3.1, lack of the references of the datasets you used.
4. The cite form is wrong in ICLR, pls refer to previous ICLR paper.

---

### Official Review · Reviewer_FYKd · 2024-11-02

**Soundness:** 2
**Presentation:** 2
**Contribution:** 2
**Rating:** 5
**Confidence:** 3

**Summary:**

This study introduces DBGMS, a novel framework for robust auditory attention decoding using EEGs. DBGMS employs a dual-branch architecture to capture temporal and frequency features,
and incorporates branch-specific GANs for high-quality data augmentation. A multi-task self-supervised learning strategy is further employed to capture generalizable attention-related features.
Evaluations on the KUL and DTU datasets demonstrate DBGMS’s superior performance across diverse experimental settings.

**Strengths:**

The paper is well-written and easy to understand, with a clear illustration of the DBGMS framework.
Experiments across subjects and datasets with varying decision window lengths show that DBGMS outperforms existing methods.
Ablation studies further confirm the effectiveness of main modules within DBGMS.

**Weaknesses:**

- The dual-branch architecture is proposed to capture more comprehensive features. However, the combination of temporal and frequency attention appears incremental,
as similar dual-branch structures have been employed to fuse temporal-frequency transformers for EEG decoding [1]. DBGMS seems to present a straightforward combination of existing temporal-frequency transformers
with graph learning. It would enhance the novelty of the work to clarify the specific differences between these approaches.

- Part of the reasons behind why DBGMS is able to extract more generalizable features is unclear.  The authors claim that multi-task self-supervised learning enhances generalization, but the content in Section 2.4 and Eq. (27) is somewhat
confusing regarding how this multi-task learning is implemented and how the tasks are selected.
Additional content would be helpful, including: 1) a detailed explanation of how the multi-task learning is implemented; 2) the rationale behind the selection of these self-supervised tasks;
and 3) empirical results demonstrating the impact of each task on generalization performance.

- Some minor points: 1) The font size in Figure 1 and Figure 2 is somewhat small. 2) The notation $\mathcal{L}_i$  is not clearly explained. 3) Citations are preferably formatted in parentheses using $\verb|\citep{}|$.

[1] Li X, Wei W, Qiu S, et al. TFF-Former: Temporal-frequency fusion transformer for zero-training decoding of two BCI tasks//Proceedings of the 30th ACM international conference on multimedia. 2022: 51-59.

**Questions:**

- The total loss illustrated in Eq. (30) appears not to include the term for the discriminator $D$. What loss function is used to train the discriminator? Additionally, could you clarify the entire training procedure of DBGMS?

- Vanilla GANs are known for their training instability. Do you employ any techniques to enhance the training stability of GANs?
   Could you provide further analysis on the training stability of DBGMS?

- Are the results shown in Figure 4 based on training and testing on the same subject? How does DBGMS perform in cross-subject scenarios? Can DBGMS achieve few-shot or zero-shot adaptation?

---

### Official Review · Reviewer_HUnR · 2024-11-04

**Soundness:** 2
**Presentation:** 1
**Contribution:** 3
**Rating:** 3
**Confidence:** 2

**Summary:**

The manuscript presents a  generative-adversarial-network-based decoding pipeline for auditory attention decoding from EEG. If I understood correctly, two autoencoder networks operating in the frequency and time domain are adversarially trained on the EEG data. Their representations/encoding are then used for downstream task (potentially with a self-supervised finetuning stage before). They present improved pefromance on auditory attention tasks compared to prior work and present ablations showing all components of their pipeline are necessary to achieve the best performance.

**Strengths:**

* Diverse set of sensible approaches combined
* Fairly large ablation (see Table 4)
* A lot of comparison baselines in table 3

**Weaknesses:**

Unfortunately, I found the manuscript in its current form extremely hard to read and understand. The overall approach was hard to understand for me, I am still not sure if I understood it correctly (see questions). I assume it is in the end an adversarial autoencoder what is trained here, but that is never stated anywhere, neither is the "adversarial autoencoder" paper cited.

The text is sometimes long and imprecise:
"An adjacency matrix A ∈ R C×C is employed to describe the intrinsic relationships between the EEG channels (nodes). The elements of this matrix are predetermined based on the spatial relationship of the EEG channels. The entry of the adjacency matrix ai,j measures the level of connection between the channels i and j."
This is quite long and still does not specificy precisely what the entries are, is it the inverse of the squared distance for example? This could be both shorter and more informational/precise at the same time.

Sometimes unnecessary terms are mentioned
"This extractor processes both generated signals in parallel, leveraging their complementary information to create a rich, multi-dimensional representation of the EEG data."
multi-dimensional seems unnecessary and vague here (encodings are typical always multidimensional, no need for different generators for that), in general multi-dimensional is used in a vague and confusing way to me in this manuscript.

Figures are not legible, see Figures 1 and 2, fonts are small and often hardly or not at all legible.

**Questions:**

So is this an adversarial autoencoder? The generator is fed a real EEG signal as an input to generate a synthetic EEG signal correct?

Which time frequency transform? Is it Fourier transform? Why not write explicitly...

In the unlabelled GAN training as well as in self-supervsed learning are the models also trained on evaluation sets of later datasets? Or how is the split during different training stages?

---

### Official Review · Reviewer_uon5 · 2024-11-04

**Soundness:** 3
**Presentation:** 2
**Contribution:** 3
**Rating:** 6
**Confidence:** 4

**Summary:**

The paper presents DBGMS (Dual-Branch Generative Adversarial Network with Multi-Task Self-Supervised Enhancement), a framework designed for robust auditory attention decoding from EEG signals. The key points/innovations of the paper include:

1. A dual-branch architecture that combines temporal attention and frequency residual learning for comprehensive EEG feature extraction.
2. Generative Adversarial Networks (GANs) for data augmentation in temporal and frequency domains.
3. Incorporation of attention mechanisms and graph convolutions for enhanced spatial-temporal feature extraction.
4. A multi-task self-supervised learning strategy using tasks like temporal order prediction and frequency band reconstruction to improve model generalization across subjects.

The framework is evaluated on two public EEG datasets (KUL and DTU) and shows improvements in detection accuracy, especially for short decision windows.

**Strengths:**

1. Innovative Architecture:
The dual-branch GAN framework is novel, leveraging temporal and spectral features.
2. Data Augmentation:
The use of GANs to generate synthetic data in both temporal and frequency domains helps mitigate the data scarcity problem.
3. Self-Supervised Learning:
The multi-task self-supervised approach effectively enhances the model’s ability to generalize across different subjects and auditory environments.
4. Comprehensive Evaluation:
The model is tested on multiple EEG datasets and shows robust performance under various conditions, including different decision window lengths.

**Weaknesses:**

1. Complexity: The proposed model introduces a high level of complexity with the dual-branch structure, GANs, and self-supervised tasks, which may pose challenges for real-time application in terms of computational efficiency. Do authors have comment on this?

2. Limited Real-World Testing: The experiments are conducted on two specific datasets, and while they show good results, the model's generalization to real-world environments with more diverse subjects and noise conditions is not fully explored.

3. Impact of Hyperparameters: The paper does not discuss the sensitivity of the model to hyperparameter tuning, especially for the GAN-based augmentation and self-supervised learning tasks, which could influence performance outcomes.

**Questions:**

In addition to weakness, please refer to these questions as well:

1. Can you provide more details on the computational efficiency of DBGMS in real-time applications? Given the dual-branch architecture and use of GANs, how does it perform in terms of training and inference time?

2. How does the model handle variability in real-world EEG data beyond the specific datasets used (KUL and DTU)? Are there plans to test the model on more diverse and noisy environments?

3. he use of GANs for augmentation is innovative, but what measures are taken to ensure that the synthetic data generated by the GANs accurately represent the underlying EEG signal distributions? There is limited statistical insight in the paper.

4. Can you clarify the interpretability of the attention mechanisms used in the model? How do you ensure that the model’s focus on specific EEG segments aligns with attention-related brain signals?

5. Although the model shows improved performance on the KUL and DTU datasets, the paper does not sufficiently address how the model performs under different noise levels in EEG signals.

---

### Note · Authors · 2024-12-03

I have read and agree with the venue's withdrawal policy on behalf of myself and my co-authors.